# Social Challenges as a Consequence of the COVID-19 Pandemic among South Florida Preschool Children with Disability and Obesity

**DOI:** 10.3390/children10040619

**Published:** 2023-03-25

**Authors:** Ruby A. Natale, Erika Paola Viana Cardenas, Parisa Varanloo, Ruixuan Ma, Yaray Agosto, Joanne Palenzuela, Julieta Hernandez, Michelle Schladant, Martha Bloyer, Sarah E. Messiah

**Affiliations:** 1Department of Pediatrics, University of Miami School of Medicine, Mailman Center for Child Development, Miami, FL 33130, USA; 2Division of Biostatistics, Department of Public Health Science, University of Miami School of Medicine, Miami, FL 33136, USA; 3Department of Physical Therapy, University of Miami, Coral Gables, FL 33146, USA; 4University of Texas Health Science Center, School of Public Health, Dallas Campus, Dallas, TX 75390, USA; 5Center for Pediatric Population Health, Children’s Health System of Texas and UT Health School of Public Health, Dallas, TX 75390, USA

**Keywords:** COVID-19, childcare industry, preschool age children, disability, obesity, BMI

## Abstract

The COVID-19 pandemic has negatively impacted the childcare industry over the past two years. This study examined how pandemic-related challenges impacted preschool-aged children by disability and obesity status. Participants were 216 children (80% Hispanic, 14% non-Hispanic Black) aged 2 to 5 years in 10 South Florida childcare centers. In November/December 2021, parents completed a COVID-19 Risk and Resiliency Questionnaire, and body mass index percentile (BMI) was collected. Multivariable logistic regression models examined the association of COVID-19 pandemic-related social challenges (transportation, employment) and child BMI and disability status. As compared to normal-weight children, those families with a child who was obese were more likely to report pandemic-related transportation (OR: 2.51, 95% CI: 1.03–6.28) challenges and food insecurity (OR: 2.56, 95% CI: 1.05–6.43). Parents of children with disabilities were less likely to report that food did not last (OR: 0.19, 95% CI: 0.07–0.48) and that they could not afford balanced meals (OR: 0.33, 95% CI: 0.13–0.85). Spanish-speaking caregivers were more likely to have a child who was obese (OR: 3.04, 95% CI: 1.19–8.52). The results suggest that COVID-19 impacts obese preschool children from Hispanic backgrounds, while disability was a protective factor.

## 1. Introduction

According to a 2021 report by the World Health Organization, over 39 million children under the age of 5 were overweight (85th ≤ body mass index percentile for age and sex) or obese (95th ≤ body mass index percentile for age and sex) [1]. Unhealthy weight is not only a major risk factor for noncommunicable diseases (i.e., cardiovascular diseases, diabetes, and hypertension), but is also associated with a higher risk of premature death and disability in adulthood [1]. According to the 2017–March 2020 National Health and Nutrition Examination Survey (NHANES), 19.7% of youth aged 2 to 19 years had obesity, and its prevalence increased with age [2]. Furthermore, the prevalence of obesity among preschoolers aged 2 to 5 years was 12.7% [2]. It is reported that children and adolescents with disabilities are at a 2–3 times greater risk of being overweight or obese than those without disabilities [3]. Children with disabilities, as described at the Convention of Rights of Person with Disabilities, are “children 18 years or younger who have ‘long-term physical, mental, intellectual, or sensory impairments which in interaction with various barriers may hinder their full and effective participation in society on an equal basis with others.” [4]. In the 2022 disability report by the United Nations Children’s Fund (UNICEF), the prevalence estimates of having moderate-to-severe disabilities globally are 28.9 million or 4.3% (95% CI: 4.1–4.6) of children aged 0–4 years, 207.4 million or 12.5% (95% CI: 11.7–13.3) of children aged 5–17 years, and 236.4 million or 10.1% (95% CI: 9.6–10.6) of all children aged 0–17 years [4]. Additionally, children with disabilities are disproportionately impacted by other health issues including asthma, breathing problems, high blood pressure, and obesity [3,5] compared to children without disabilities.

According to Shoesmith and colleagues, educational settings such as childcare centers are ideal locations to implement continuous and intensive health interventions targeting key risk factors [6]. However, the COVID-19 pandemic has resulted in negative health and social impacts to children and their families worldwide [7]. Rundle et al. noted that school closures may exacerbate the epidemic of childhood obesity and increase disparities in obesity risk [8]. Indeed, studies have shown that there has been an overall increase in BMI in children during the COVID-19 pandemic [9,10,11,12,13]. The COVID-19 pandemic has also highlighted racial and ethnic disparities across many domains, including in child health [14]. Children at higher risk for poor COVID-19-related outcomes include those with underlying medical conditions, those of African American or Non-Hispanic Black descent [15], and those from low-income backgrounds who face food insecurity [16]. Families of children with disabilities were also found to have higher rates of work loss and financial burden [17,18,19]; and a lack of access to needed therapies, medical supplies, and nursing care [20]. 

Little is known regarding the impacts of COVID-19 on families with young children aged 2 to 5 years who have either a disability or obesity or both. The purpose of the current study was to determine the association of COVID-19-related negative social challenges, including parents’ job status (unemployment due to the pandemic), transportation challenges, and food insecurity outcomes in the target population. It was hypothesized that families with a child with a disability would be disproportionately impacted by COVID-19-related social challenges versus families who did not have a child with a disability. It was also hypothesized that families with a child who is obese would be disproportionately impacted by COVID-19-related social challenges as compared to families with a child who is normal weight.

## 2. Materials and Methods

### 2.1. Participants and Procedures

Participants were recruited from ten early childcare centers (CCCs) serving low resource families in Miami-Dade County. Parents consented to participate as part of a larger randomized controlled trial. The larger study was designed to examine the impacts of an obesity prevention program on child BMI. Ten childcare centers in Miami-Dade County, Florida, were recruited to participate based on the criteria that they had at least 30 children enrolled and that at least 20% of the children had a disability. Five centers were chosen to receive the obesity prevention program, called Healthy Caregivers–Healthy Children (HC2), which consisted of 24 lesson plans to improve the nutrition and physical activity environment of the childcare center. Five centers received a mental health program which consisted of 24 lesson plans to improve the social–emotional environment of the childcare centers. Once the center director committed to participate, research assistants recruited parents at drop-off and pick-up times at the centers to obtain consent and complete baseline survey packets. Participants were given the option to complete consent forms and surveys based on their primary language. Consent forms and surveys were available in English and Spanish, and if literacy issues were present, the research assistant was able to read the items to the parents.

Consent forms and baseline survey packets were distributed during November–December 2021. Once consent forms were obtained, the research assistants returned to the centers to collect each child’s height and weight in accordance with CDC standards. Research assistants attended a one-week training on the research protocol and methods to reduce bias related to disability and obesity, including role playing scenarios and on-site training. The training was conducted by a multidisciplinary team including a staff member who was a mental health counselor and who had a disability, a clinician with a background in special education, a clinical child psychologist with expertise in obesity prevention in childcare, and a physical therapist with a background in special needs. A detailed protocol regarding implementation in childcare centers was followed. Fidelity was also assessed by the program manager at an initial and then 1-month follow-up to ensure the protocol was followed. A total of 216 parents consented to having their children ages 2–5 years participate in this study and completed baseline surveys. This study was approved by the University of Miami Institutional Review Board (Clinical Trial # NCT05106426). 

### 2.2. Measures

HC2 Caregiver Interview to Document Disability Status and Food Insecurity. Our HC2 baseline questionnaire was used to capture age, sex, race/ethnicity (children and parents). Caregivers were also asked to identify their child’s disability status and developmental milestones using the following five self-report questions. (1) Does this child have a documented disability? (2) Did a teacher, doctor, or other health professional ever tell you they have a concern about this child’s learning, development, or behavior? (3) Do you have any concerns about this child’s learning, development, or behavior? (4) Are you concerned about how this child: (a) talks and makes speech sounds? (b) understands what you say? (c) uses (his/her) hands and fingers to do things? (d) uses (his/her) arms and legs? (e) behaves? (f) gets along with others? (g) is learning to do things for (himself/herself)? (h) is learning pre-school or school skills? (5) Has this child ever been referred to Early Steps or Florida Diagnostic and Learning Resources System (FDLRS) or Miami-Dade County Public Schools for testing or evaluation? If a parent answered “Yes” to any of these five questions, their child was considered to be a child with special health care needs. Household food insecurity was assessed using two items from the US Household Food Security Survey Module [21]: “The food that we bought just didn’t last, and we didn’t have money to get more” and “We couldn’t afford to eat balanced meals (meals containing 3 different food groups)”.

Anthropometry. Height (stadiometer) and weight (digital scale) were collected for each child and converted to an age- and sex-adjusted BMI percentile. Data collection methods were based on US Department of Health and Human Services (US HHS) guidelines [22]. 

COVID-19 Risk and Resilience. Since no COVID-19 measure existed at the beginning of the pandemic, the Everyday Stressors Index [23], which was developed in 1988, was modified by the authors to include stressors as a result of COVID-19. This measure is still relevant today as a measure of stress [24,25]. Parents indicated their level of concern at baseline for their health, the health of family members, employment, housing, transportation, having enough money for basic necessities, and relationships. Respondents indicated their level of concern along a Likert scale ranging from (1) not at all bothered, (2) a little bothered, (3) somewhat bothered, (4) bothered a great deal, or (0) do not know. The Everyday Stressors Index demonstrates good reliability, validity, and internal consistency, including for samples of low-income families with young children [24]. Internal consistency for the adapted scale in our sample was high: ɑ = 0.87.

### 2.3. Sample Size Calculation 

This analysis consisted of baseline data from a larger cluster randomized controlled trial. Clusters were considered to be childcare centers as the primary unit of analysis. Thus, this is an additional individual-level post hoc analysis. As this is a retrospective analysis using our baseline data, we performed a post hoc power analysis and found that our study had ample (>99.9%) statistical power to detect differences in obesity status (12.9% of the population^3^) and disability status (>10% of the population^4^) at the alpha level of 0.05. The pandemic provided a unique opportunity to capture data at the individual level to examine how children and their families were coping with COVID-19-related challenges. Information learned here can help inform future public health strategies for families that shared the information collected in this supplemental study.

### 2.4. Statistical Analyses

Children’s BMI was treated as BMI percentile adjusted for age and sex, according to the World Health Organization (WHO) growth standard charts to monitor the growth of children aged from birth to younger than 2 years [26], CDC 2000 growth reference charts to monitor the growth of children aged 2 to 20 years without obesity [27], and CDC 2022 extended BMI-for-age growth charts for children and adolescents aged 2 to 20 years with obesity [28,29,30]. Children’s BMI percentile for age and sex was converted to categorical variables. Normal weight was greater than the 5th BMI percentile for age and sex and less than the 85th. Overweight was equal to or greater than the 85th BMI percentile for age and sex and less than the 95th percentile. Obese was considered equal to or greater than the 95th BMI percentile for age and sex.

Univariate analysis was performed to summarize the participants’ characteristics, such as demographics, food insecurity, and COVID-19-related outcomes in count and percentage. Bivariate analysis was then conducted by using Pearson’s chi-squared test and Fisher’s exact test, as appropriate, to examine the relationship between participants’ characteristics and COVID-19-related outcomes and children’s disability status (Disability/Non-Disability), and with children’s BMI percentile for age and sex (3 levels as Normal Weight, Overweight, Obese). The count and percentage, along with the corresponding *p*-value, are reported in Table 1 and Table 2, respectively.

Multivariable logistic regression models were implemented to test the effect of children’s disability and COVID-19 negative social challenges (Job problem, Transportation problem, food insecurity, etc.), as well as caregivers’ country of origin and language preference, on the prevalence of children’s obesity, which was treated as two binary variables (children with Overweight versus children with Normal Weight, and children with Obese versus children with Normal Weight). The children’s race/ethnicity and children’s gender were adjusted in the multivariable logistic regression models described above as covariates. The estimated odds ratio, 95% confidence interval, and *p*-value can be found in Table 3. There were three multivariable logistic regression models that were also conducted to test the effect of children’s BMI percentile categories (3-level variable with Normal weight as reference group) and children’s disability status on COVID-19-related binary outcomes, such as Job problem due to COVID-19 (Worried versus Not At All Worried), Transportation problem due to COVID-19 (Worried versus Not At All Worried), Food did not last but no money to get more food (True versus Never True), and Cannot afford balanced meal (True versus Never True). Children’s race/ethnicity, gender, immigration status, and English proficiency were also adjusted as confounders in the models. The estimated odds ratio and 95% confidence interval, and the corresponding *p*-value for each logistic regression can be found in Table 4. A *p*-value less than 0.05 was considered statistically significant. Data were analyzed using R studio version 4.0.3.

## 3. Results

The final analytical sample consisted of 216 children. Table 1 shows demographic factors related to disability status, but given that 36 participants did not complete the disability status questions, the sample in Table 1 is a total of 180 participants (49% female, 80% Hispanic, 6% non-Hispanic White, 14% non-Hispanic Black, 51% with a disability). The majority (52%) of participants were normal weight, while 17% were overweight, and 31% were obese. The majority (63%) of children with disabilities were normal weight, while among those who were obese, 60% did not have a disability. Similarly, 64% of those with severe obesity did not have a disability. There was a high prevalence of food insecurity among participant families; 37% reported that they did not have enough money to purchase food, and 39% reported they could not afford a balanced meal. Two thirds (65%) of those who reported food insecurity had a child with no disability. Most families (63%) reported no transportation challenges due to COVID-19, but 56% of families were worried about COVID-19-related employment challenges (Table 1). Fifty-nine percent of caregivers were born outside of the US, 53% of whom had been living at least 10 years in the United State of America. Spanish was the primary language of 53% of parents. Of the parents, 62% were fluent in English.

Disability status was significantly associated with ethnicity (*p*-value: 0.03), BMI (*p*-value: 0.24), and food insecurity (*p*-value < 0.001). Specifically, the prevalence of disability was significantly higher among Hispanics (78%) compared to the other ethnicities (22%). Children with a disability had a higher prevalence of normal BMI (63%) compared to those children with no disability (42%). Furthermore, parents of children with no disability had a significantly higher prevalence of food insecurity compared to those with a disability (food did not last: 50% versus 24%, *p*-value < 0.001, and families could not afford balanced meals: 50% versus 29%, *p*-value = 0.01) (Table 1).

The prevalence of normal weight was higher among children with disability (61%). In addition, the prevalence of children with normal weight was higher among families who did not have food insecurity (enough money to get food: 74%; could afford a balanced meal: 72%), any transportation issues (72%), or any job problem due to COVID-19 (51%). Conversely, the prevalence of obesity was significantly higher among Hispanic children. Moreover, the prevalence of child obesity was higher among families who were not fluent in English (62%). Additionally, families for whom Spanish was their primary language had a higher prevalence of child obesity (75%). There was no statistically significant difference for the sex and ethnicity between children’s BMI percentile categories (Table 2).

Multivariable logistic regression analyses showed that children with disability were approximately 50% less likely to have obesity (OR 0.44, CI: 0.19–0.97, *p*-value = 0.045) compared to children with no disability. Moreover, families who were worried about transportation due to COVID-19 were over two times as likely to have a child with obesity versus parents who were not worried about transportation barriers (OR 2.51, 95% CI: 1.03–6.28, *p*-value = 0.045). Parents who had food insecurity were almost two times as likely to have a child who was overweight versus parents with no food insecurity (OR 2.90, 95% CI: 1.08–8.10, *p*-value = 0.04). Similarly, parents who reported not being able to afford balanced meals were over three times as likely to have a child who was overweight (OR 3.43, 95% CI: 1.28–9.68, *p*-value = 0.02) and over two times as likely to have a child who was obese (OR 2.56, 95% CI: 1.05–6.43 *p*-value = 0.04) versus parents who could afford balanced meals. Families who were fluent in English were more than 60% less likely to have a child who was obese (OR 0.37, 95% CI: 0.17–0.80 *p*-value = 0.01). Conversely, families whose primary language was Spanish were over three times more likely to have a child who was obese (OR 3.04, 95% CI: 1.19–8.52, *p*-value = 0.03) (Table 3).

Figure 1 shows two models with an adjusted odds ratios for primary predictors of children who were overweight. Model A shows children who were obese. Model B shows the odds ratios after multiple variables were adjusted, controlling for child age, child gender, and child race/ethnicity.

When BMI was treated as a predictor in multivariable regression analysis, results showed that families who had a child with a disability had lower odds of food insecurity (food did not last: OR 0.19, 95% CI 0.07–0.48, *p*-value < 0.001, and cannot afford balanced meal: OR 0.33, 95% CI 0.13–0.85, *p*-value = 0.02) compared to families who did not have a child with a disability. Parents who were born overseas were over nine times more likely to not have enough money to get more food (OR 9.34, 95% CI 2.42–45.0, *p*-value = 0.002) and over five times more likely to not be able to afford a balanced meal (OR 5.76, 95% CI 1.64–24.0, *p*-value = 0.01) compared to parents who were born in the United States. Seven (11%) US-born participants reported not having enough money to get more food compared to forty-eight (52%) participants who were born overseas. Ten (15%) US-born participants reported not being able to afford a balanced meal compared to fifty (55%) participants who were born overseas. Parents who were fluent in English were approximately 80% less likely to have any transportation issues due to COVID-19 (OR 0.22, 95% CI 0.07–0.64, *p*-value = 0.01) and more than 60% less likely to have a problem with affording a balanced meal (OR 0.34, 95% CI 0.12–0.95, *p*-value = 0.04) compared to parents who are not proficient in English (Table 4).

## 4. Discussion

Recent research on the consequences of the COVID-19 pandemic on the social challenges of young children with disabilities and/or unhealthy weights indicates a relationship between disruptions in daily routines and interruptions in special education interventions and an exacerbated risk for obesity, respectively [31,32]. The findings of the current study suggest pandemic-related impacts on the health of Hispanic preschool children at the intersection of obesity and disability.

The results suggest that there are more significant pandemic impacts on transportation challenges and food insecurity for families of children who are obese. In contrast, families of children with disabilities were less likely to report pandemic impacts, especially as it relates to food insecurity. Overall, those who spoke Spanish were more likely to have a child who was obese. Those who were born outside of the US were more likely to have COVID-19-related food insecurity. These results can inform the development of prevention efforts within a young and vulnerable population at risk for the development of obesity.

Study findings suggest that parents who have a preschooler who is obese are facing many challenges (transportation, food insecurity). As children’s level of obesity increases, so do the challenges for these families [33]. A study of 432,302 children aged from 2 to 19 years found the rate of body mass index (BMI) increase nearly doubled during the COVID-19 pandemic compared to a pre-pandemic period. This faster increase was most pronounced in children with overweight or obesity and younger school-aged children [31,32]. In the current study, families who were worried about transportation due to COVID-19 were over 2.5 times as likely to have a child who was obese. Similarly, parents who were worried about food insecurity due to COVID-19 were almost 2.5 times as likely to have a child who was obese. One explanation is that those who are concerned about transportation may have limited funds and, thus, buy the least expensive foods that are less nutritious. The literature is clear that cheaper foods are unhealthy and highly processed but control hunger [34], making cheaper foods a common choice in food-insecure households [35]. Chain supply problems during the pandemic and access to food markets with healthier food options in certain neighborhoods could also explain limited food choices and overall food scarcity [36].

These findings related to obesity expose some of the social determinant challenges experienced by Hispanic families with young children in particular and in complex ways. This finding suggests that system-level solutions may be important in addressing how to assist families already experiencing multiple challenges. One example may be a multi-level intervention to not only help families with vocational planning and transportation but also assess and address how these factors impact children in their care.

This study also suggests that children with a disability had a higher prevalence of normal BMI compared to children with no disability. In contrast to our hypothesis, the results suggest that disability status may have been a protective factor during COVID-19. In addition, the previous literature has shown a higher prevalence of obesity in children with disabilities [37], which contrasts with study results that showed that those with a disability are half as likely to be impacted by obesity. In addition, some of the research that was conducted during COVID-19 suggests the highest increases in BMI were found among those who were underweight, with no significant increase in those who were overweight or obese [10,38]. Moreover, contrary to our prediction, children with no disability had a higher prevalence of food insecurity compared to children with disability. One explanation for this finding may be based on research that shows that families with a child with a disability may have parental resiliency factors that may buffer impacts [39]. One study conducted with parents of children with disabilities during COVID-19 examined parental resiliency defined as parents’ perception of self, their plans for the future, and levels of family cohesion [39]. They found that perception of self, planned future, and family cohesion served as protective factors that partially buffered the impacts of COVID-19 [39].

Our study findings further suggest that the prevalence of child obesity was higher among families who were not fluent in English. Additionally, families for whom Spanish was their primary language had a higher prevalence of child obesity. The literature has consistently shown that Hispanic children have higher rates of obesity [40,41]. During the pandemic, several studies showed an increase in obesity [42,43,44] and specifically among Hispanics [45]. Beyond ethnicity, this study also showed that those who were born outside of the US had higher rates of food insecurity. One explanation for this finding may be that immigrant families may have limited access to food stamps, the supplemental nutrition assistance program—SNAP (only for those family members born in the US), or may not qualify at all. Immigrant Hispanic families may also not qualify for Medicaid to access primary and specialized pediatric care. Data were not collected regarding if the families were illegal immigrants and the impacts of that on the outcomes. This is an area that researchers may consider exploring.

Another factor to consider regarding overweight and obesity found in Hispanics is that overweight in infants may be a result of maternal excessive stress [46,47]. While maternal stress was not measured in this study, future research may consider exploring this variable and the possibility of counseling to decrease maternal stress.

It should also be noted that the majority of our sample was Hispanic (approximately 80%) as compared to Non-Hispanic Black (14%) and only 6% Non-Hispanic White; therefore, the reference group was Hispanics. The Non-Hispanic White sample were not significantly different from the Hispanic population; however, this may be an artifact of the small and unbalanced sample sizes.

### Study Limitations and Strengths

The majority of this sample consisted of children who were Hispanic or Non-Hispanic Black (94%). While this is a strength of the study given the limited amount of the literature available regarding obesity, ethnicity, and childhood disability during the COVID-19 pandemic, the results should not be generalized to other populations. In addition, the sample sizes for the varying ethnicities were not balanced. In particular, only having one non-Hispanic White subject limits the accuracy of chi-square analyses. It should also be noted that although this study was designed to be a cross-sectional analysis to show COVID-19 impacts, 59% of our caregivers were not born in the US and, thus, may not have had job security pre-pandemic, which could not be accounted for in the analysis.

In addition, data were not collected on the severity of disability for each child. It is plausible that the severity of disability may impact BMI, and so it is suggested that future research include this as a variable to study. It should also be noted that these data were collected in Miami-Dade County from October to December 2021, during which time the county was considered to be a COVID-19 hot spot as the Delta variant was predominant. Data regarding if the child had COVID-19 were not collected, and the impact of variants on BMI has yet to be studied, especially among young children. Furthermore, we do not have data on the BMI of this sample prior to COVID-19, and thus, we cannot make any causal inferences regarding the impacts of COVID-19 on BMI. Although this is a cross-sectional snapshot of what was occurring in that timeframe, the findings can still help to elucidate the impact of COVID-19 on the weight status of children with and without disabilities.

## 5. Conclusions

COVID-19 has impacted the growth and development of young children and their families. It is important to properly assess the impacts of COVID-19 on child body mass index (BMI), especially within vulnerable populations such as children with disabilities. With an increase in obesity among pediatric populations, it is imperative to invest in prevention strategies that will contribute to improved health and functioning for this population. In particular, the results suggest the need for strategies to screen and treat obesity within preschool populations. In addition, it may be beneficial to examine ways to decrease the stress of those Hispanic families of overweight and obese children. Research in this area can further be expanded by assessing the social determinant challenges experienced by these families, thus addressing the long-term impacts of COVID-19.

## Figures and Tables

**Figure 1 children-10-00619-f001:**
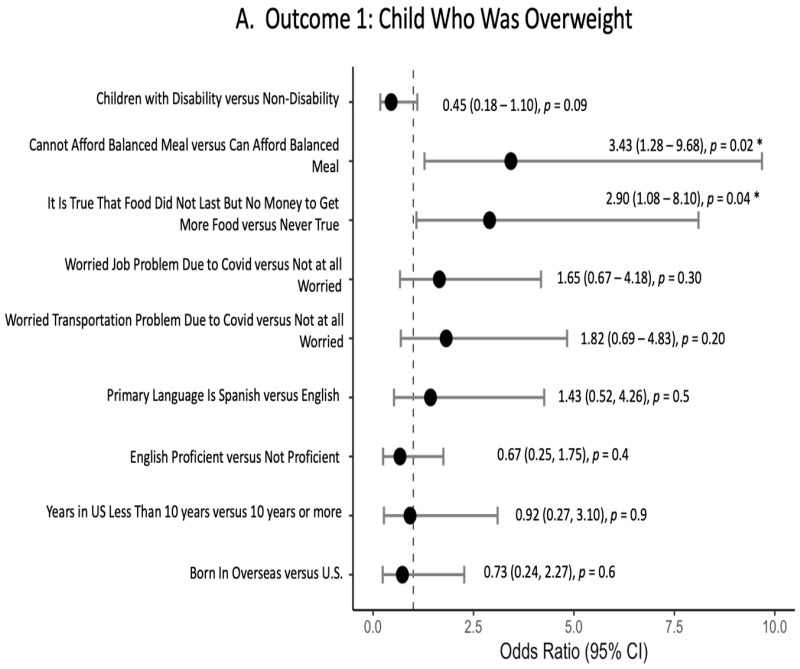
Adjusted odds ratio with 95% confidence interval for primary predictors of (**A**) children who are overweight and (**B**) children who are obese after multivariable adjustment, controlling for child age, child gender, and child race/ethnicity. * Indicates statistical significance.

**Table 1 children-10-00619-t001:** Participant Descriptive and COVID-19 Pandemic-Related-Challenges by Disability Status.

	Overall N = 216 **	Non-Disability N = 89	Disability N = 91	*p*-Value
**Sex**				0.4
Female	82 (49)	44 (52)	38 (45)	
Male	87 (51)	41 (48)	46 (55)	
**Ethnicity**				0.03 *
Hispanic	121 (80)	65 (82)	56 (78)	
Non-Hispanic White	9 (6)	1 (1)	8 (11)	
Non-Hispanic Black	21(14)	13 (17)	8 (11)	
**Child’s BMI Percentiles Status**				0.02 *
Normal BMI	94 (52)	37 (42)	57 (63)	
Overweight	31 (17)	19 (21)	12 (13)	
Obese	55 (31)	33 (37)	22 (24)	
**Food Did Not Last But No Money to Get More Food**				<0.001 *
Never True	103 (63)	40 (50.0)	63 (76)	
True	60 (37)	40 (50.0)	20 (24)	
**Cannot Afford Balanced Meal**				0.01 *
Never True	99 (61)	40 (50)	59 (71)	
True	64 (39)	40 (50)	24 (29)	
**Transportation Due to COVID-19**				0.10
Not at all worried	99 (63)	49 (58)	50 (70)	
Worried	57 (37)	36 (42)	21 (30)	
**Job Problem Due to COVID-19**				0.2
Not at all worried	72 (44)	32 (39)	40 (49)	
Worried	92(56)	51 (61)	41 (51)	
**Country of Birth of the Caregiver**				0.6
United States	70 (41)	35 (43)	35 (39)	
Overseas	100 (59)	46 (57)	54 (61)	
**Years living in the US**				0.2
Less than 10 years	49 (47)	27 (53)	22 (42)	
10 years or more	55 (53)	24 (47)	31 (58)	
**English Proficiency**				0.02 *
English proficient	106 (62)	45 (53)	61 (71)	
Non-English Proficient	65 (38)	40 (47)	25 (29)	
**Primary Language**				0.2
English	72 (42)	33 (39)	39 (45)	
Spanish	91 (53)	50 (59)	41 (48)	
Creole	3 (2)	0 (0)	3 (3.5)	
Other	5 (3)	2 (2)	3 (3.5)	

* Statistically significant; ** 36 missing data responses about disability status.

**Table 2 children-10-00619-t002:** Participant Descriptive and COVID-19 Pandemic-Related Challenges by Weight Status.

	Overall N = 216	Normal Weight N = 113	Overweight N = 35	Obese N = 68	*p*-Value
**Sex**					0.8
Female	91 (48)	48 (50)	16 (47)	27 (45)	
Male	99 (52)	48 (50)	18 (53)	33 (55)	
**Ethnicity**					0.04 *
Hispanic	141 (82)	61 (76)	26 (76)	54 (93)	
Non-Hispanic White	9 (5)	7 (9)	2 (6)	0 (0)	
Non-Hispanic Black	22 (13)	12 (15)	6 (18)	4 (7)	
Other	0 (0.0)	0 (0.0)	0 (0.0)	0 (0.0)	
**Children’s Disability Status**					0.02 *
Non-Disability	89 (49)	37 (39)	19 (61)	33 (60)	
Disability	91 (51)	57 (61)	12 (39)	22 (40)	
**Food Did Not Last But No Money to Get More Food**					0.02 *
Never True	103 (63)	62 (74)	16 (53)	25 (51)	
True	60 (37)	22 (26)	14 (47)	24 (49)	
**Cannot Afford Balanced Meal**					0.01 *
Never True	99 (61)	60 (72)	15 (50)	24 (48)	
True	64 (39)	23 (28)	15 (50)	26 (52)	
**Transportation Problem Due to COVID-19**					0.04 *
Not at all worried	99 (63)	58 (72)	18 (60)	23 (50)	
Worried	57 (37)	22 (28)	12 (40)	23 (50)	
**Job Problem Due to COVID-19**					0.2
Not at all worried	72 (44)	43 (51)	11 (37)	18 (36)	
Worried	92 (56)	41 (49)	19 (63)	32 (64)	
**Caregiver Country of Origin**					0.2
United States	70 (41)	39 (44)	14 (50)	17 (32)	
Overseas	100 (59)	50 (56)	14 (50)	36 (68)	
**Years living in the US**					>0.9
Less than 10 years	49 (47)	23 (46)	8 (50)	18 (47)	
10 years or more	55 (53)	27 (54)	8 (50)	20 (53)	
**English Proficiency**					<0.001 *
English proficient	110 (57)	66 (68)	20 (59)	24 (38)	
Non-English Proficient	84 (43)	31 (32)	14 (41)	39 (62)	
**Primary Language**					0.003 *
English	76 (39)	49 (51)	14 (41)	13 (21)	
Spanish	110 (57)	44 (45)	19 (56)	47 (75)	
Creole	3 (1)	1 (1)	0 (0)	2 (3)	
Other	5 (3)	3 (3)	1 (3)	1 (1)	

* Statistically significant.

**Table 3 children-10-00619-t003:** Odds of Abnormal Weight by Various Descriptive and Food Insecurity Variables.

		Outcome 1 (Child Who Was Overweight) b		Outcome 2 (Child Who Was OBESE) c	
Independent Variable	Category	Odds Ratio (95% CI)	*p*-Value	Odds Ratio (95% CI)	*p*-Value
**Children with Disability**	No	Reference		Reference	
	Yes	0.45 (0.18–1.10)	0.09	0.44 (0.19–0.97)	0.045 *
**Transportation Due to COVID-19**	Not at all worried	Reference		Reference	
	Worried	1.82 (0.69–4.83)	0.2	2.51 (1.03–6.28)	0.045 *
**Job Problem Due to COVID-19**	Not at all worried	Reference		Reference	
	Worried	1.65 (0.67–4.18)	0.3	1.71 (0.74–4.03)	0.2
**Food Did Not Last But No Money to Get More Food**	Never True	Reference		Reference	
	TRUE	2.90 (1.08–8.10)	0.04 *	2.24 (0.94–5.47)	0.07
**Cannot Afford Balanced Meal**	Never True	Reference		Reference	
	TRUE	3.43 (1.28–9.68)	0.02 *	2.56 (1.05–6.43)	0.04 *
**Country of Origin**	United States	Reference		Reference	
	Overseas	0.73 (0.24, 2.27)	0.6	1.41 (0.55, 3.73)	0.5
**Years in United States**	10 or more years	Reference		Reference	
	Less than 10 years	0.92 (0.27, 3.10)	0.9	0.80 (0.31, 2.05)	0.6
**English Proficiency**	Not proficient	Reference		Reference	
	Proficient	0.67 (0.25, 1.75)	0.4	0.37 (0.17, 0.80)	0.01 *
**Primary Language**	English	Reference		Reference	
	Spanish	1.43 (0.52, 4.26)	0.5	3.04 (1.19, 8.52)	0.03 *

“*” Indicates *p*-value < 0.05, which is considered as statistically significance. b—Overweight (≥85th and <95th) versus Normal (≥5th and <85th). c—Obese (≥95th) versus Normal (≥5th and <85th).

**Table 4 children-10-00619-t004:** Multivariable Logistic Regression Models to Compare COVID-19-Related Outcomes in Child BMI Percentile Categories, Controlling for Disability Status, Child Ethnicity, Child Gender, Country of Origin, and English Proficiency.

	Model 1 Compare Job Problem Due to COVID-19 ^c^		Model 2 Compare Transportation Problem Due to COVID-19 ^c^		Model 3 Compare Food Did Not Last But No Money to Get More Food ^d^		Model 4 Compare Cannot Afford Balanced Meal ^d^	
Independent Variable	Odds Ratio (95% CI)	*p*-Value ^a^	Odds Ratio (95% CI)	*p*-Value ^a^	Odds Ratio (95% CI)	*p*-Value ^a^	Odds Ratio (95% CI)	*p*-Value ^a^
**Child BMI percentile Categories ^b^**								
Child who was Normal Weight	Reference		Reference		Reference		Reference	
Child who was Overweight	1.51 (0.58–4.09)	0.40	1.34 (0.44–4.00)	0.60	1.90 (0.57–6.46)	0.30	2.66 (0.80–9.30)	0.11
Child who was Obese	1.61 (0.67–3.96)	0.30	2.32 (0.85–6.49)	0.10	1.24 (0.43–3.50)	0.70	1.42 (0.50–3.95)	0.50
**Child who has Disability**								
No	Reference		Reference		Reference		Reference	
Yes	1.01 (0.46–2.27)	0.90	1.16 (0.45–3.01)	0.80	0.19 (0.07–0.48)	**<0.001 ***	0.33 (0.13–0.85)	**0.02 ***
**Child Ethnicity**								
Hispanic	Reference		Reference		Reference		Reference	
Non-Hispanic White	0.67 (0.08–4.11)	0.70	0.68 (0.03–5.25)	0.70	1.83 (0.08–17.5)	0.60	1.19 (0.06–10.2)	0.90
Non-Hispanic Black	1.21 (0.35–4.15)	0.80	0.40 (0.05–1.93)	0.30	1.16 (0.19–6.64)	0.90	0.54 (0.07–3.11)	0.50
**Child Gender**								
Female	Reference		Reference		Reference		Reference	
Male	1.57 (0.74–3.33)	0.20	2.06 (0.87–5.02)	0.10	1.12 (0.45–2.79)	0.80	1.23 (0.50–3.07)	0.70
**Country of Origin**								
United States	Reference		Reference		Reference		Reference	
Overseas	1.61 (0.59–4.44)	0.40	0.72 (0.21–2.38)	0.60	9.34 (2.42–45.0)	**0.002 ***	5.76 (1.64–24.0)	**0.01 ***
**English Proficiency**								
Not proficient	Reference		Reference		Reference		Reference	
Proficient	0.50 (0.19–1.31)	0.20	0.22 (0.07–0.64)	**0.01 ***	0.51 (0.17–1.47)	0.20	0.34 (0.12–0.95)	**0.04 ***

^a^ “*” Indicates *p*-value < 0.05, which is considered statistically significant. ^b^ Normal defined as child BMI percentile ≥5th and <85th. Overweight defined as child BMI percentile ≥85th and <95th. Obese defined as child BMI percentile ≥95th. ^c^ Worried versus Not at all worried (reference). ^d^ True versus Never True (reference).

## Data Availability

Given the confidentiality of the subjects, data are not available to be shared.

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
