# Peer review of "Social Challenges as a Consequence of the COVID-19 Pandemic among South Florida Preschool Children with Disability and Obesity"

_children, 2023, doi:10.3390/children10040619_

Round 1

Reviewer 1 Report

Thank you for the opportunity to review this very interesting study, which addresses a major health concern in children. I agree with your study design and methods, and I agree with your conclusions, based on study results. I have a couple of questions and suggestions relating to your presentation of study results:

1. I cannot get the same percentages you report in Tables 1 and 2. For example, in Table 1, I get the percentage of female subjects as 45.6% (82/180). I am finding this repeatedly in both Tables.

2. Please include a post-hoc analysis (adjusted residuals, for example) for your Chi square results in Tables 1 and 2 that involve more than 2 categories (ethnicity and BMI categories). This allows the reader to see where the differences lie. For example, in Table 1 the P value for disability status versus BMI percentile status is 0.04. With 8 comparisons at an alpha of 0.05, an adjusted residual of +/- 2.73437 or greater is significant. The only significant adjusted residual (+/- 2.829) is between disabled and non-disabled children with a normal BMI. For children with severe obesity (the focus of your results and discussion), the adjusted residual is only +/- 1.56865, so this comparison was not significant. 

3. Please discuss a small sample size as a study limitation, as this likely affected your study results and adjusted residuals. In particular, only having 1 non-disabled non-Hispanic White subject limits the accuracy of Chi square, as this violates the basic assumptions of Chi square.

Author Response

Reviewer 1

Comment: Thank you for the opportunity to review this very interesting study, which addresses a major health concern in children. I agree with your study design and methods, and I agree with your conclusions, based on study results. I have a couple of questions and suggestions relating to your presentation of study results.

Response: Thank you for the positive review and thoughtful comments. We appreciate your suggestions as noted below.

Comment 1: I cannot get the same percentages you report in Tables 1 and 2. For example, in Table 1, I get the percentage of female subjects as 45.6% (82/180). I am finding this repeatedly in both Tables.

Response: The percentages are according to the data available, therefore, the discrepancy in percentages you noted may be due to variables that have missing data. For instance, 82 participants are women among the169 participants (both gender) from whom we have information about disability; that explains the 49% (82/169). A footnote has been added to Table 1 and Table 2 explaining the discrepancies.

Comment 2. Please include a post-hoc analysis (adjusted residuals, for example) for your Chi square results in Tables 1 and 2 that involve more than 2 categories (ethnicity and BMI categories). This allows the reader to see where the differences lie. For example, in Table 1 the P value for disability status versus BMI percentile status is 0.04. With 8 comparisons at an alpha of 0.05, an adjusted residual of +/- 2.73437 or greater is significant. The only significant adjusted residual (+/- 2.829) is between disabled and non-disabled children with a normal BMI. For children with severe obesity (the focus of your results and discussion), the adjusted residual is only +/- 1.56865, so this comparison was not significant. 

Response: We have added a post-analysis in the results and in Table 1. “Once the significant correlation was detected for the variables, the post hoc analysis based on the residuals of Pearson’s Chi-squared test for count data was then subsequently performed, for the variables with the number of categories greater than two, to examine where the significant differences lie in the pairwise comparisons, with the p-value adjusted by the Bonferroni correction method”. For Table 2 the ORs in Table 3 reflect the pairwise relationship and the significance is consistent with the post hoc analysis results.

Comment 3. Please discuss a small sample size as a study limitation, as this likely affected your study results and adjusted residuals. In particular, only having 1 non-disabled non-Hispanic White subject limits the accuracy of Chi square, as this violates the basic assumptions of Chi square.

Response: We have added a statement to the limitations regarding the small sample size. “In addition, the sample sizes for the varying ethnicities were not balanced. In particular, only having one non-Hispanic White subject limits the accuracy of Chi square analyses.”

Reviewer 2 Report

This is an insightful study that provides some interesting ideas for future research. Please see my comments and suggestions below to improve the quality of the manuscript.

Line 25: Should be written as follows: “….more likely to report pandemic related transportation (OR: 4.58, 95% CI: 1.32 - 21.4)…”

Lines 25-27: Were these analyses conducted comparing severe obese versus normal weight children? If yes, then you need to make it clearer and position the statistics in the appropriate places.

Line 36: “…age of 5 years are overweight…”

Line 42: “…ages 2-to-19 years..”

Line 47: “..aged 3-to-17 years…”

Line 68: “…ages 2-to-5 years….” – please add ‘years’ after the last digit when reporting age throughout the manuscript.

Line 111: “Anthropometry”

Line 115: “COVID-19 Risk and Resilience…” This should be a subheading and not part of the sentence as seen through no full stop.

Lines 125-126: This sentence is very poorly written. Please revise.

Lines 134-137: All these percentiles need a range (i.e., lower and higher thresholds) otherwise they make no sense.

Line 171: “…17% were overweight…”

Line 195: “…74%..”

Line 200: “….There was no statistically….”

Line 213: “…child who was overweight…”

Line 215: “…child who was overweight…” also in Figure 1 (Line 231)and Table 4 category column change the wording  to “child who was overweight”

Line 252: “…COVID-19…”

Line 274: Do you mean “…regarding severe obesity…”

Lines 282-285: This sentence is difficult to understand. Please revise.

Line 286: “…but also assess and address how these factors impact children in their care.”

Line 289: “…compared to children with no disability….”

Line 301: “….that families with a disabled child…”

Line 335: “….impact of COVID-19 on weight status of children with….”

Line 343: “….for strategies to screen and treat severe obesity….”

Line 345: “…social determinant challenges…”

Author Response

Reviewer 2

Thank you for the detailed comments. We appreciate your time and believe that your comments have helped us significantly improve the manuscript.

Comment: Line 25: Should be written as follows: “….more likely to report pandemic related transportation (OR: 4.58, 95% CI: 1.32 - 21.4)…”

Reply: This has been corrected.

Comment: Lines 25-27: Were these analyses conducted comparing severe obese versus normal weight children? If yes, then you need to make it clearer and position the statistics in the appropriate places.

Reply: Yes, this is current. The sentences were rewritten to compare analyses to those who were normal weight.

Comment: Line 36: “…age of 5 years are overweight…”

Reply: This has been corrected.

Comment: Line 42: “…ages 2-to-19 years..”

Reply: This has been corrected.

Comment: Line 47: “..aged 3-to-17 years…”

Reply: This has been corrected.

Comment: Line 68: “…ages 2-to-5 years….” – please add ‘years’ after the last digit when reporting age throughout the manuscript.

Reply: Noted.

Comment: Line 111: “Anthropometry”

Reply: Corrected.

Comment: Line 115: “COVID-19 Risk and Resilience…” This should be a subheading and not part of the sentence as seen through no full stop.

Reply: Corrected

Comment: Lines 125-126: This sentence is very poorly written. Please revise.

Reply: This analysis consisted of baseline data from a larger cluster randomized controlled trial. Clusters were considered to be childcare centers as the primary unit of analyses.

Comment: Lines 134-137: All these percentiles need a range (i.e., lower and higher thresholds) otherwise they make no sense.

Reply: All of the ranges for the BMI percentiles have been added.

Comment: Line 171: “…17% were overweight…”

Reply: This has been changed.

Comment: Line 195: “…74%..”

Reply: This has been changed.

Comment: Line 200: “….There was no statistically….”

Reply: This has been changed.

Comment: Line 213: “…child who was overweight…”

Reply: This has been changed.

Comment: Line 215: “…child who was overweight…” also in Figure 1 (Line 231)and Table 4 category column change the wording  to “child who was overweight”

Reply: This has been changed.

Comment: Line 252: “…COVID-19…”

Reply: This sentence has been changed.

Comment: Line 274: Do you mean “…regarding severe obesity…”

Reply: The discussion has been rewritten.

Comment: Lines 282-285: This sentence is difficult to understand. Please revise.

Reply: This sentence has been rewritten “This finding suggests that system-level solutions  may be important in addressing how to assist families already experiencing multiple challenges.”

Comment: Line 286: “…but also assess and address how these factors impact children in their care.”

Reply: This sentence has been rewritten as suggested.

Comment: Line 289: “…compared to children with no disability….”

Reply:  This has been changed.

Comment: Line 301: “….that families with a disabled child…”

Reply: This sentence has been revised to be consistent with people’s first language.

Comment: Line 335: “….impact of COVID-19 on weight status of children with….”

Reply: This has been changed.

Comment: Line 343: “….for strategies to screen and treat severe obesity….”

Reply: This has been changed.

Comment: Line 345: “…social determinant challenges…”

Reply: This has been changed.

Reviewer 3 Report

This is a report of an individual-level post-hoc analysis of a larger cluster randomized controlled trial to determine the social challenges of obese and disabled Florida preschoolers as a consequence of the COVID-19 pandemic. Conducted in the Miami-Dade County, the study included both parents and children. The majority of the participants were Hispanic, some Black and few White. The purpose was to determine the association of COVID-19-related negative social challenges including job status (unemployment due to the pandemic), transportation challenges, and food insecurity outcomes of this population. The hypothesized was that families having a child with a disability would be disproportionately impacted by COVID-19 related social challenges versus families who did not have a child with a disability. This hypothesis was not found to hold true, with the results suggesting that disability in these preschool children may have been a protective factor during COVID-19.

The strengths of this paper are that it is clearly and logically written. On the surface, this seems like an important and well-done study. However, if examined closely, a number of problems can be noted. The first is that the study is a post-hoc analysis from a larger study. This larger study is likely the one cited as reference 19, which, to date, has not been published. Given that readers are asked to refer to reference 19 to find out the details of this larger study, and this is currently not possible given that it is not published, this is a major problem. Generally, the authors cite and argue in relation to research that is out of date. This is irrelevant to their work as cutting-edge researchers—updated references must be found and their work considered in relation to these current references. As well, the authors want to point to race as being a significant factor in this study. However, since they did not include if they were immigrants, whether they had the ability to understand English, if they had job security before the pandemic, or if they were illegal aliens, these important considerations may be of more relevance than race in and of itself. Maybe the most problematic thing about this work is that it is based on the premise that no work has been done in this area before. If the authors check current research since 2020, they will find this to be untrue—there seems to be quite a lot of research that has now been conducted.

In all, this study can still be valuable, but the authors have to be much more careful about what they claim and having current research to support their positions. As well, the limitations section must be expanded in the manner mentioned above. Furthermore, readers cannot be asked to reference a paper that has not been published to find out details of the method. These details must be added to this paper.

Please review the information regarding the reference style used by the journal on the supplied Word template. The style of referencing that the authors have used does not conform to MDPI reference style. The references will need to be reformatted in the correct style. All sections, subsections and subsubsections should be numbered and the numbering must follow the MDPI style guide found on the template. 

Line by line suggested edits

4 “Preschool Children by Disability and Weight Status”—please move this information to line 3 to be part of the title line.  Please change the title to conform more to the purpose of the study and its result. The suggestion is “Social Challenges of Obese and Disabled Florida Preschoolers as a Consequence of the COVID-19 Pandemic”

21 Change “childcare centers” to “Florida childcare centers”.

33 Please arrange the keywords so they follow how they appear in the abstract. As well, COVID-19 and childcare industry should be added as a keyword and early childhood does not appear in the Abstract, so it should not be part of the keywords—better to say preschool age children, which does appear. Change “obesity; early childhood; preschool; BMI; disability” to “COVID-19; childcare industry; preschool age children; disability; obesity; BMI”.

36 Change “Over 39 million children under the age of 5 had” to “ According to a 2021 report by the World Health Organization, over 39 million children under the age of 5 were at that time”.

44 Please provide a definition of disability supported by a current peer reviewed reference.

45 Citation 3 is to a reference from 2015. Please find a supporting reference in a peer reviewed journal from the last five years.

46-52 The work of these authors is reporting a study in 2023; therefore, the background information in the Introduction must be from research published as close in time to 2023 as possible. What follows is a link to a Google Scholar search that will provide current research in this regard—please rewrite this paragraph to report on more current research than is currently cited in lines 46-52.

World Health Organization. WHO European regional obesity report 2022. World Health Organization. Regional Office for Europe. 2022. Available online: https://apps.who.int/iris/bitstream/handle/10665/353747/9789289057738-eng.pdf?sequence=1&isAllowed=y (accessed on 8 February 2023).

65 Given that reference 17 is from 2009 it can have nothing to say about problems regarding work loss and financial burden during COVID-19. Please delete this reference and add one from after January 2020.

70 Change “including job” to “including parents’ job”.

77 change “were consented” to “consented”.

79 From the reference provided to citation 19, it appears that this research has not been published. As such, if it has not been published, it cannot be cited as a location to find the details about the methods and procedures. Either complete the reference showing the article is in fact published or provide the missing details about the methods and procedures in this paper.

82 Please provide information on how the consent forms were obtained from the participants.

84 Please provide information on what was the focus of the training on the research protocol as well as the methods used to reduce bias. Were the methods particularly concerning disability and obesity? Please explain.

85 Please indicate why there were more parents than children and state the number of children who had both parents consent to the study and the number who had only one parent.

89 Please write out HC2 in full before using the acronym.

92-94 As there were only five different questions, rather than providing examples, please list them all.

97 Please either explain why the 2012 survey module was used as opposed to a more recent one.

114 The Guidelines used are out of date. There are more recent ones from 2015-2018. Please redo your analysis based on these updated guidelines: https://stacks.cdc.gov/view/cdc/100478

115 If reference 22 is to the seminal work regarding the Everyday Stressors Index, please mention that this index was developed in 1988. However, the authors also have to provides current references to peer reviewed research to indicate that this Index is still relevant today.

116 Who modified the Index to include stressors as a result of COVID-19? If it was the authors, please say so.

123 Reference 23 is too old to provide a current demonstration of these variables. Please find a current reference (within the last five years).

125 Please fix the formatting of the indentation to correspond with other indentation for first lines of paragraphs.

126-127 Please explain more regarding what the authors mean by “school was the primary unit of analysis”. There needs to be additional information on this randomized controlled trial. Why was the decision made to extract this “individual-level post-hoc analysis”? Who made this decision?

166 Please explain why the final sample consisted by 216 rather than the original 233 children.

181-184 Table 1 does not conform to the guidelines on the MDPI Word template for how tables and their footnotes are to be constructed. Please refer to the guidelines and redo the table.

203-205  Table 2 does not conform to the guidelines on the MDPI Word template for how tables and their footnotes are to be constructed. Please refer to the guidelines and redo the table.

219-225 Although Table 3 is constructed more exactly to the MDPI guidelines than either Table 1 or Table 2, the Category and P-value columns do not line up. As well, the Odds Ratio (95% CI) column under Outcome 2 needs to split Odds Ratio to one line and (95% CI) to the line below, similar to the Outcome 1 and Outcome 3 column. Furthermore, the authors need to eliminate the heavy line separating the title of the outcomes from the titles below them and use a regular weight line. Make sure that the information under Reference in each of the relevant columns follows directly on the line below it, not with a line space. This will require making all columns with Reference the same width. 

228-233 Figure 1 is not mentioned in the text of the paper. It must be mentioned to be included. As well, please reformat this figure so that Outcome 2 is under Outcome 1 and all the outcome figures are lined up. The Outcome 2 figure current is cut off at the right edge of the document and can’t be seen.

237-244 Although Table 4 is constructed more exactly to the MDPI guidelines than either Table 1 or Table 2, the authors need to eliminate the heavy line separating the title of the model numbers from the titles below them and use a regular weight line. Make sure that the information under Reference in each of the relevant columns follows directly on the line below it, not with a line space. Table 4 needs to follow mention of the table in the text. Therefore, it should follow line 253.

257 Although the authors make the claim that there has been no research to “address the intersection of severe obesity and disability in the early childhood years (ages 2-to-5)”, the following link to a Google Scholar search demonstrates that, since 2020, there have been a number of articles written on this subject. Please read through these articles and indicate how the research conducted in this current study by the authors differs from these already published studies. https://scholar.google.ca/scholar?as_ylo=2019&q=intersection+of+severe+obesity+and+disability+in+the+early+childhood+years+(ages+2-to-5)&hl=en&as_sdt=0,5

267-268 “As children’s level of obesity increases”—need a reference to support this claim.

275 The reference is from 2017. Now that it is 2023, it would be better to find an updated reference to support this claim.

277-278 Need a reference to support that there were chain supply problems during the pandemic. Need another reference to support that there was less access to heathier food options in certain neighborhoods.

289 “which is consistent with the literature”—please cite this literature.

292 Reference 27 is far too old to be relevant. Please find an updated reference.

294 “One explanation”—another explanation is that the cited reference is over twelve years old. This section of the paragraph considering reasons for a discrepancy related to outdated research needs to be redone once the authors find a current reference.

301-302 Please cite the research to support this hypothesis.

315-321 The authors cannot both say that 80% of the sample were Hispanic and that the majority of the sample “were children from diverse ethnic” backgrounds. In what way were the backgrounds diverse? Please explain.

336 Another major limitation of this study eluded to in line 310 is that, rather than the race being the determining factor, it is that a good proportion of the Hispanic families are immigrants. This may mean they don’t speak English and, therefore, may not have access to information provided only in English. As well, they may have had insecure jobs even before the pandemic—some may be illegal immigrants. As a result, without this additional information, there can be no conclusion drawn regarding the importance of race with respect to obesity and disability concerning COVID-19 limitations.

355-363 Please reduce the font size of the headings of each of these sections.

364 All references must be redone to conform to MDPI style, including reducing the font size.

Author Response

Reviewer #3

Comment: The strengths of this paper are that it is clearly and logically written. On the surface, this seems like an important and well-done study. However, if examined closely, a number of problems can be noted. The first is that the study is a post-hoc analysis from a larger study. This larger study is likely the one cited as reference 19, which, to date, has not been published. Given that readers are asked to refer to reference 19 to find out the details of this larger study, and this is currently not possible given that it is not published, this is a major problem.

Reply: In effort to meet the strict word limitations we thought it best to reference the paper, however, we have now added this information as you suggested.

Comment: Generally, the authors cite and argue in relation to research that is out of date. This is irrelevant to their work as cutting-edge researchers—updated references must be found and their work considered in relation to these current references.

Reply: References have been updated as suggested below. All 45 references are now between 2020 and 2023 with one reference from 2018 because it is based on Bronfenbrenner’s  ecological mobile, one from 2019 based on BMI measurement, and one from 2015 because it shows similar results based on our previous work with a similar population and one is from 1988 because it is the original Everyday Stressors references but we added two additional references from 2021 to show how this scale is still applicable today..

Comment: As well, the authors want to point to race as being a significant factor in this study. However, since they did not include if they were immigrants, whether they had the ability to understand English, if they had job security before the pandemic, or if they were illegal aliens, these important considerations may be of more relevance than race in and of itself.

Reply: Thank you for the comment. We re-ran the analyses to add country of origin, years living in the US, and primary language in the models to account for these other variables as you suggested. We modified the results, statement in the discussion and in the limitations. It should be noted that the measures were all translated into Spanish and a Research Assistant was available to read the question items in their native language if literacy was an issue. This was added to the methods section.

Comment:  Maybe the most problematic thing about this work is that it is based on the premise that no work has been done in this area before. If the authors check current research since 2020, they will find this to be untrue—there seems to be quite a lot of research that has now been conducted. In all, this study can still be valuable, but the authors have to be much more careful about what they claim and having current research to support their positions.

Reply: We have updated several of the references as described that are current between 2020-2023. We also modified the discussion.

Comment: As well, the limitations section must be expanded in the manner mentioned above.

Reply: Thank you the comments. The limitations section has been updated to include these suggestions.

Comment: Furthermore, readers cannot be asked to reference a paper that has not been published to find out details of the method. These details must be added to this paper.

Reply: Yes, we agree. As stated above these details have now been added.

Comment: Please review the information regarding the reference style used by the journal on the supplied Word template. The style of referencing that the authors have used does not conform to MDPI reference style. The references will need to be reformatted in the correct style. All sections, subsections and subsubsections should be numbered and the numbering must follow the MDPI style guide found on the template.

Reply: References have been updated.

Line by line suggested edits

Comment: 4 “Preschool Children by Disability and Weight Status”—please move this information to line 3 to be part of the title line.  Please change the title to conform more to the purpose of the study and its result. The suggestion is “Social Challenges of Obese and Disabled Florida Preschoolers as a Consequence of the COVID-19 Pandemic”

Reply: Based on your comment, and in alignment with People’s First Language, we modified the title to “Social Challenges as a Consequence of the COVID-19 Pandemic Among Florida Preschool Children with Disability and Obesity

Comment: 21 Change “childcare centers” to “Florida childcare centers”.

Reply: Thank you for the suggestion. This was added.

Comment: 33 Please arrange the keywords so they follow how they appear in the abstract. As well, COVID-19 and childcare industry should be added as a keyword and early childhood does not appear in the Abstract, so it should not be part of the keywords—better to say preschool age children, which does appear. Change “obesity; early childhood; preschool; BMI; disability” to “COVID-19; childcare industry; preschool age children; disability; obesity; BMI”.

Reply: Thank you for the suggestion. The key words were changed.

Comment: 36 Change “Over 39 million children under the age of 5 had” to “ According to a 2021 report by the World Health Organization, over 39 million children under the age of 5 were at that time”.

Reply: This sentence has been revised.

Comment: 44 Please provide a definition of disability supported by a current peer reviewed reference.

Reply: This has been added to the introduction. Children with disabilities, as described at the Convention of Rights of Person with Disabilities, are “children 18 years or younger who have ‘long-term physical, mental, intellectual, or sensory impairments which in interaction with various barriers may hinder their full and effective participation in society on an equal basis with others.” [4]. 

Comment 45: Citation 3 is to a reference from 2015. Please find a supporting reference in a peer reviewed journal from the last five years.

Reply: Citation 3 was replaced.
Walker, M., Nixon, S., Haines, J., & McPherson, A. C. (2019). Examining risk factors for overweight and obesity in children with disabilities: A commentary on Bronfenbrenner’s ecological systems framework. Developmental neurorehabilitation22(5), 359-364.

Comment: lines 46-52 The work of these authors is reporting a study in 2023; therefore, the background information in the Introduction must be from research published as close in time to 2023 as possible. What follows is a link to a Google Scholar search that will provide current research in this regard—please rewrite this paragraph to report on more current research than is currently cited in lines 46-52.

World Health Organization. WHO European regional obesity report 2022. World Health Organization. Regional Office for Europe. 2022. Available online: https://apps.who.int/iris/bitstream/handle/10665/353747/9789289057738eng.pdf?sequence=1&isAllowed=y (accessed on 8 February 2023).

Reply: Thank you for the comment. We agree and we have updated introduction with new statements and references. For example, “It is reported that children and adolescents with disabilities are 2-3 times at greater risk to be overweight or obese than those without disabilities [3].  Children with disabilities, as described at the Convention of Rights of Person with Disabilities, are “children 18 years or younger who have ‘long-term physical, mental, intellectual, or sensory impairments which in interaction with various barriers may hinder their full and effective participation in society on an equal basis with others.” [4].  In the 2022 disability report by United Nations Children’s Fund (UNICEF), the prevalence estimates of having moderate-to-severe disabilities globally are; 28.9 million or 4.3% (95% CI: 4.1–4.6) of children aged 0–4 years, 207.4 million or 12.5% (95% CI: 11.7–13.3) of children aged 5–17 years, and 236.4 million or 10.1% (95% CI: 9.6–10.6) of all children aged 0–17 years. [4]. Additionally, children with disabilities are disproportionately impacted by other health issues including asthma, breathing problems, high blood pressure, and obesity [3 ,5] compared to children without disabilities.”

Comment: Given that reference 17 is from 2009 it can have nothing to say about problems regarding work loss and financial burden during COVID-19. Please delete this reference and add one from after January 2020.

Reply: We have replaced reference 17 with three new references that support the statement regarding the impacts of COVID-19.

Neece C, McIntyre LL, Fenning R. Examining the impact of COVID19 in ethnically diverse families with young children with intellectual and developmental disabilities. Journal of Intellectual Disability Research [Internet]. 2020 Oct 1 [cited 2023 Feb 21];64(10):739. Available from: /pmc/articles/PMC7461180/

Iovino EA, Caemmerer J, Chafouleas SM. Psychological distress and burden among family caregivers of children with and without developmental disabilities six months into the COVID-19 pandemic. Res Dev Disabil [Internet]. 2021 Jul 1 [cited 2023 Feb 21];114:103983. Available from: /pmc/articles/PMC9758884/

Nicholas DB, Mitchell W, Ciesielski J, Khan A, Lach L. A Qualitative Examination of the Impact of the COVID-19 Pandemic on Individuals with Neuro-developmental Disabilities and their Families. J Child Fam Stud [Internet]. 2022 Aug 1 [cited 2023 Feb 21];31(8):2202–14. Available from: https://link.springer.com/article/10.1007/s10826-022-02336-8

Comment: 70 Change “including job” to “including parents’ job”.

Reply: This has been changed.

Comment: 77 change “were consented” to “consented”.

Reply: This has been changed.

Comment: 79 From the reference provided to citation 19, it appears that this research has not been published. As such, if it has not been published, it cannot be cited as a location to find the details about the methods and procedures. Either complete the reference showing the article is in fact published or provide the missing details about the methods and procedures in this paper.

Reply: As stated above, this information has been added.

Comment: 82 Please provide information on how the consent forms were obtained from the participants.

Reply: This information has been added.

Comment: 84 Please provide information on what was the focus of the training on the research protocol as well as the methods used to reduce bias. Were the methods particularly concerning disability and obesity? Please explain.

Reply: This information has been added.

Comment: 85 Please indicate why there were more parents than children and state the number of children who had both parents consent to the study and the number who had only one parent.

Reply: The 269 includes parents who consented but did not complete any survey items or did not provide sufficient information to calculate the child’s BMI. Therefore, we have revised this statement to include only those who consented with completed surveys to avoid confusion.

Comment: 89 Please write out HC2 in full before using the acronym.

Reply: Healthy Caregivers-Healthy Children (HC-2) was written out in the previous paragraph.

Comment: 92-94 As there were only five different questions, rather than providing examples, please list them all.

Reply: These questions have been included.

Comment: 97 Please either explain why the 2012 survey module was used as opposed to a more recent one.

Reply: This survey was part of a larger protocol in which the 2012 survey was used.

Comment: 114 The Guidelines used are out of date. There are more recent ones from 2015-2018. Please redo your analysis based on these updated guidelines: https://stacks.cdc.gov/view/cdc/100478

Reply: The analyses were redone using these guidelines as suggested and new tables and figures were included.

Comment: 115 If reference 22 is to the seminal work regarding the Everyday Stressors Index, please mention that this index was developed in 1988. However, the authors also have to provides current references to peer reviewed research to indicate that this Index is still relevant today.

Reply:  Thank you for the comment. This statement has been modified “COVID-19 Risk and Resilience was measured by the Everyday Stressors Index [22] which was developed in 1988, which was modified by the authors to include stressors as a result of COVID-19 since no COVID-19 measure existed at the beginning of the pandemic.”. In addition, references were updated to support this statement.

Comment: 116 Who modified the Index to include stressors as a result of COVID-19? If it was the authors, please say so.

Reply: See above comment.                     

Comment: 123 Reference 23 is too old to provide a current demonstration of these variables. Please find a current reference (within the last five years).

Reply: This reference was updated.

Davidson B, Schmidt E, Mallar C, Mahmoud F, Rothenberg W, Hernandez J, et al. Risk and resilience of well-being in caregivers of young children in response to the COVID-19 pandemic. Transl Behav Med [Internet]. 2021 Feb 1 [cited 2023 Feb 21];11(2):305–13. Available from: /pmc/articles/PMC7890655/

Comment: 125 Please fix the formatting of the indentation to correspond with other indentation for first lines of paragraphs.

Reply:  Thank you noticing. This has been fixed. However, due to the extensive changes that have made using track changes, as requested, the formatting may appear to be off until the track changes has been accepted.

Comment: 126-127 Please explain more regarding what the authors mean by “school was the primary unit of analysis”. There needs to be additional information on this randomized controlled trial. Why was the decision made to extract this “individual-level post-hoc analysis”? Who made this decision?

Reply: The pandemic provided a unique opportunity to capture data at the individual level to examine how children and their families were coping with COVID-19 related challenges. Information learned here can help inform future public health strategies for families that shared the information collected in this supplemental study. This certainly could not have been foreseen when the original cluster RCT was funded so the PI made the decision to add COVID-19 questions. Please also remember that all anthropometric, disability and other data is collected at the individual level too.

Comment: 166 Please explain why the final sample consisted by 216 rather than the original 233 children.

Reply: As stated above the sample of 216 were those who consented and had a survey item completed.

Comment: 181-184 Table 1 does not conform to the guidelines on the MDPI Word template for how tables and their footnotes are to be constructed. Please refer to the guidelines and redo the table.

Reply: This table has been reformatted. However, please note that since track changes was required for the resubmission, the formatting may be off until the track changes are accepted.

Comment: 203-205  Table 2 does not conform to the guidelines on the MDPI Word template for how tables and their footnotes are to be constructed. Please refer to the guidelines and redo the table.

Reply: This table has been reformatted. However, please note that since track changes was required for the resubmission, the formatting may be off until the track changes are accepted.

Comment: 219-225 Although Table 3 is constructed more exactly to the MDPI guidelines than either Table 1 or Table 2, the Category and P-value columns do not line up. As well, the Odds Ratio (95% CI) column under Outcome 2 needs to split Odds Ratio to one line and (95% CI) to the line below, similar to the Outcome 1 and Outcome 3 column. Furthermore, the authors need to eliminate the heavy line separating the title of the outcomes from the titles below them and use a regular weight line. Make sure that the information under Reference in each of the relevant columns follows directly on the line below it, not with a line space. This will require making all columns with Reference the same width.

Reply: This table has been reformatted. However, please note that since track changes was required for the resubmission, the formatting may be off until the track changes are accepted.

Comment: 228-233 Figure 1 is not mentioned in the text of the paper. It must be mentioned to be included. As well, please reformat this figure so that Outcome 2 is under Outcome 1 and all the outcome figures are lined up. The Outcome 2 figure current is cut off at the right edge of the document and can’t be seen.

Reply:  This statement was added “Figure 1 shows 2 models with adjusted odds ratios for primary predictors of children with overweight. Model A shows children with obesity. Model B shows the odds ratios after multiple variables were adjusted, controlling child age, child gender, and child race/ethnicity.”

Comment: 237-244 Although Table 4 is constructed more exactly to the MDPI guidelines than either Table 1 or Table 2, the authors need to eliminate the heavy line separating the title of the model numbers from the titles below them and use a regular weight line. Make sure that the information under Reference in each of the relevant columns follows directly on the line below it, not with a line space. Table 4 needs to follow mention of the table in the text. Therefore, it should follow line 253.

Reply: This table has been reformatted. However, please note that since track changes was required for the resubmission, the formatting may be off until the track changes are accepted.

Comment: 257 Although the authors make the claim that there has been no research to “address the intersection of severe obesity and disability in the early childhood years (ages 2-to-5)”, the following link to a Google Scholar search demonstrates that, since 2020, there have been a number of articles written on this subject. Please read through these articles and indicate how the research conducted in this current study by the authors differs from these already published studies. https://scholar.google.ca/scholar?as_ylo=2019&q=intersection+of+severe+obesity+and+disability+in+the+early+childhood+years+(ages+2-to-5)&hl=en&as_sdt=0,5

Reply: Thank you for the link. We have reviewed the published studies and as a result we rewrote the first paragraph of the Discussion to reflect the research you provided and what our research adds to this.

Comment: 267-268 “As children’s level of obesity increases”—need a reference to support this claim.

Reply: We have added a reference to support this claim.
Centers for Disease Control and Prevention. Erratum: Vol.70, No. 37. Morbidity and Mortality Weekly Report (MMWR) [Internet]. 2021 Sep 24 [cited 2023 Feb 21];70(38):1355. Available from: https://www.cdc.gov/mmwr/volumes/70/wr/mm7038a6.htm

Comment: 275 The reference is from 2017. Now that it is 2023, it would be better to find an updated reference to support this claim.

            Reply: This reference was updated. Fan L, Canales E, Fountain B, Buys D. An Assessment of the Food Retail Environment in Counties with High Obesity Rates in Mississippi. https://doi.org/101080/1932024820201852147 [Internet]. 2020 [cited 2023 Feb 21];16(4):571–93. Available from: https://www.tandfonline.com/doi/abs/10.1080/19320248.2020.1852147

Comment: 277-278 Need a reference to support that there were chain supply problems during the pandemic. Need another reference to support that there was less access to heathier food options in certain neighborhoods.

Reply: This reference has been added. Aday S, Aday MS. Impact of COVID-19 on the food supply chain. Food Quality and Safety [Internet]. 2020 Dec 18 [cited 2023 Feb 21];4(4):167–80. Available from: https://academic.oup.com/fqs/article/4/4/167/5896496

Comment: 289 “which is consistent with the literature”—please cite this literature.

Reply: This reference has been added.

Masi A, Mendoza Diaz A, Tully L, Azim SI, Woolfenden S, Efron D, et al. Impact of the COVID19 pandemic on the wellbeing of children with neurodevelopmental disabilities and their parents. J Paediatr Child Health [Internet]. 2021 May 1 [cited 2023 Feb 21];57(5):631. Available from: /pmc/articles/PMC8014782/

Comment: 292 Reference 27 is far too old to be relevant. Please find an updated reference.

Reply: This reference was used.

Masi A, Mendoza Diaz A, Tully L, Azim SI, Woolfenden S, Efron D, et al. Impact of the COVID19 pandemic on the wellbeing of children with neurodevelopmental disabilities and their parents. J Paediatr Child Health [Internet]. 2021 May 1 [cited 2023 Feb 21];57(5):631. Available from: /pmc/articles/PMC8014782/

Comment: 294 “One explanation”—another explanation is that the cited reference is over twelve years old. This section of the paragraph considering reasons for a discrepancy related to outdated research needs to be redone once the authors find a current reference.

Reply: Thank you for the comment. This sentence was removed and the section was revised.

Comment: 301-302 Please cite the research to support this hypothesis.

Reply: This reference was added. Montirosso R, Mascheroni E, Guida E, Piazza C, Sali ME, Molteni M, et al. Stress Symptoms and Resilience Factors in Children With Neurodevelopmental Disabilities and Their Parents During the COVID-19 Pandemic. Health Psychology. 2021;40(7):428–38.

Comment: 315-321 The authors cannot both say that 80% of the sample were Hispanic and that the majority of the sample “were children from diverse ethnic” backgrounds. In what way were the backgrounds diverse? Please explain.

Reply: Yes, we agree that this is confusing and we have revised the comment to remove the term “diverse”.

Comment: 336 Another major limitation of this study eluded to in line 310 is that, rather than the race being the determining factor, it is that a good proportion of the Hispanic families are immigrants. This may mean they don’t speak English and, therefore, may not have access to information provided only in English. As well, they may have had insecure jobs even before the pandemic—some may be illegal immigrants. As a result, without this additional information, there can be no conclusion drawn regarding the importance of race with respect to obesity and disability concerning COVID-19 limitations.

Reply:  Thank you for the suggestion. We have re-run the models to include immigration via country of origin as well as primary language so that these variables in addition to race could be included. The results, tables, and discussion has been updated to include the new results. It should be noted that all information was provided in the family’s primary language.  

Comment: 355-363 Please reduce the font size of the headings of each of these sections.

Reply: We have reduced font size but given that we are required to use track changes not all changes will be apparent until the edits have been accepted.

Comment: 364 All references must be redone to conform to MDPI style, including reducing the font size.

Reply: Thank you for the comment. All references have been reformatted.

Round 2

Reviewer 1 Report

Thank you for addressing my comments. I feel that the revised version presents the information in a much clearer manner for the audience. This manuscript is a pleasure to read!

Author Response

Thank you!

Reviewer 2 Report

Well done on sufficiently addressing my comments and improving the quality of your manuscript.

Author Response

Thank you!

Reviewer 3 Report

This paper and the analysis of the data have been much improved over the previous version of this manuscript. As well, the references are now up to date and almost all claims, as a result, have at least one supporting reference.

Some of the English remains a problem and although the reference style at this point corresponds more closely to the MDPI standard upheld by Children, the references still need work to make them in the correct style. Similarly, although the tables are much improved, they still require a little work and the Figures need to be enlarged.

Three things are more substantial regarding work to be done. One is that the some of the statistical reasoning needs to be checked. It may be correct, but something seems off in some aspects. The authors mention resiliency at one point but they have not discussed resiliency in the paper. There needs to be a discussion about resiliency and references to peer reviewed journals regarding it. Furthermore, something else the authors have not discussed is the effect of maternal stress on weight gain of their young children during COVID-19. Maternal stress during COVID-19 is a major problem with respect to weight gain in their children and it is not mentioned by the authors. Stress needs to be included in this assessment as should some suggestions on how that stress might be reduced (counselling is a suggested method).

Page by page suggested edits.

Page 1

Change “transportation (OR: 2.51, 95% CI: 1.03 – 6.28) and food insecurity (OR: 2.56, 95% CI: 1.05– 6.43)) challenges” to “transportation (OR: 2.51, 95% CI: 1.03 – 6.28) challenges and food insecurity (OR: 2.56, 95% CI: 1.05– 6.43)”.

There are two periods after “challenges”. Please remove one.

Change ”on the health of vulnerable preschool” to “obese preschool”.

Page 2

Change “had obesity and prevalence increased” to “had obesity with its prevalence increased”.

“It was hypothesized that families with a child with a disability would be disproportionately impacted by COVID-19 related social challenges versus families who did not have a child with a disability”—what was the hypothesis regarding obesity in pre-school aged children? This paper is also on obesity. There needs to by a hypothesis related to obesity.

Page 6

Although the Table 1 has been improved, please check the Children template for tables. Notice that tables are to have only horizontal lines as dividers. There should be no vertical lines as a border or to divide columns. As well, ensure that the size of the columns are such that the titles “Overall” and “Disability” are both on one line only. Increasing their size will also ensure that the data are all only on one line a well. Under the Non-Disability column for Ethnicity, the percentages add up to 99% rather than 100%. Please change it so it adds up to 100%.

Page 7

Change “families whom Spanish” to “families for whom Spanish”.

Change “nostatistically” to “no statistically”.

Pages 7-9

Although the Table 1 has been improved, please check the Children template for tables. Notice that tables are to have only horizontal lines as dividers. There should be no vertical lines as a border or to divide columns. As well, ensure that the size of the table is such that the title “Overweight” is on one line only. To have the English correspond with the other titles, change “Obesity” to “Obese”.

Page 9

“Also, families who were worried about transportation due to COVID-19 were over 7 times as likely to have a child with obesity versus parents who were not worried about transportation barriers”—this doesn’t seem right. Please check this result.

Pages 9-10

Table 3 is done in the correct format according to the Children template except that the line above the subheading should be solid, not broken by columns. 

Page 10

The footnotes for Table 3 are repeated. Eliminate one set of footnotes.

“Figure 1 shows 2 models with”—please increase the font size of these words to correspond with the rest of the main text.

Page 11

Both Figure 1A and Figure 1B are far too small to read the text. Their size needs to be increased so that the text is legible.

Change the title of Figure 1B from “Outcome2: Child Who Was Obesity” to “Outcome 2: Child Who Was Obese”.

Page 12

Change “families who had not” to “families who didn’t have”.

“Parents who were born overseas were over 9 times more likely to not have enough money to get more food (OR 9.34, 95% CI 2.42-45.0, p- value=0.002) and over 5 times more likely to could not be able to afford balanced meal”—9 times and 5 times seem too large, is this really right?

Change “more likely to could not be able to afford” to “more likely to not be able to afford”.

Page 13

Table 4 is done in the correct format according to the Children template except that the line above the subheading should be solid, not broken by columns. 

Change, in column 1 “Child who was Obesity” to “Child who was Obese”.

Page 14

Both the first and second paragraphs of the Discussion have periods at the beginning of their first sentences. Please remove these.

Change “ethnically minoritized” to “Hispanic”. Hispanic families are not in a minority in Miami.

Change “areobese” to “are obese”.

Change “Results suggest that families of children who areobese had more significant pandemic impacts on transportation and food insecurity challenges” to “Results suggest that there are more significant pandemic impacts on transportation challenges and food insecurity for families of children who are obese.”

Change “a preschooler who areobesity” to “a preschooler who is obese”.

Change “[31,32].In” to “[31,32]. In”.

Change “food market that with” to “food markets with”.

Change “diverse racial/ethnic” to “Hispanic”.

Page 15

“One explanation for this finding is that families with a child with a disability may have parental resiliency factors that may buffer impacts”—need a current reference with respect to resiliency for this explanation.

Change “.. Study” to “Study”.

Change “that prevalence” to “that the prevalence”.

Change “families whom” to “families for whom”.

Change “and specifically amoug Hispanics in general” to “specifically among Hispanics”.

Change “that who were” to “that those who were”.

Change “stamps -supplemental” to “stamps, supplemental”.

One thing that is not mentioned regarding the overweight and obesity found in Hispanics that were studied is that overweight in infants may be a result of maternal excessive stress. Please add current references to stress and weight gain and suggestions regarding the possibility of counselling to decrease maternal stress. See, for example:

Carroll, N.; Sadowski, A.; Laila, A.; Hruska, V.; Nixon, M.; Ma, D.W.L.; Haines, J.; on behalf of the Guelph Family Health Study. The Impact of COVID-19 on Health Behavior, Stress, Financial and Food Security among Middle to High Income Canadian Families with Young Children. Nutrients 202012, 2352. https://doi.org/10.3390/nu12082352

Jansen, E.; Thapaliya, G.; Aghababian, A.; Sadler, J.; Smith, K.; Carnell, S. Parental stress, food parenting practices and child snack intake during the COVID-19 pandemic. Appetite 2021161, 105119. https://doi.org/10.1016/j.appet.2021.105119

Please delete the period before the final paragraph preceding Study Limitations and Strengths.

Change “ethnic and racial backgrounds specifically Hispanic or Non-Hispanic Black” to “Hispanic or Non-Hispanic Black”

Change “data was” to “data were” (this error appears in the final paragraph of the page twice).

Page 16

In the Conclusions, recommendations should also be made regarding ways to decrease the stress of those Hispanic families of overweight and obese children.

Please reduce the font size of “Institutional Review Board Statement”, “Informed Consent Statement”, “Data Availability Statement: Given the confidentiality of subjects, data is not available to be shared.”, “Conflicts of Interest” and “References”.

Although the style of the references is much closer to MDPI style, none of the necessary italics or bolding has been done. Please fix all the references to include italics and bolding. As well, note that it should say “Available online:” and after the address, it should say “(accessed on Day Month written as a word and Year)”.

Author Response

Reviewer 3

Comment: This paper and the analysis of the data have been much improved over the previous version of this manuscript. As well, the references are now up to date and almost all claims, as a result, have at least one supporting reference.

Response: Thank you for the comments and for your suggestions. We believe that the paper has been greatly improved as a result of your comments.

Comment: Some of the English remains a problem and although the reference style at this point corresponds more closely to the MDPI standard upheld by Children, the references still need work to make them in the correct style. Similarly, although the tables are much improved, they still require a little work and the Figures need to be enlarged.

Response: We have reviewed the references and made changes. The tables and figure were revised.

Comment: Three things are more substantial regarding work to be done. One is that the some of the statistical reasoning needs to be checked. It may be correct, but something seems off in some aspects.

Response: Our biostatistician re-ran the statistics and notes were made below.

Comment: The authors mention resiliency at one point but they have not discussed resiliency in the paper. There needs to be a discussion about resiliency and references to peer reviewed journals regarding it. Furthermore, something else the authors have not discussed is the effect of maternal stress on weight gain of their young children during COVID-19. Maternal stress during COVID-19 is a major problem with respect to weight gain in their children and it is not mentioned by the authors. Stress needs to be included in this assessment as should some suggestions on how that stress might be reduced (counselling is a suggested method).

Response: Thank you for these suggestions. As noted below, this information was added.

Page by page suggested edits.

Page 1

Comment: Change “transportation (OR: 2.51, 95% CI: 1.03 – 6.28) and food insecurity (OR: 2.56, 95% CI: 1.05– 6.43)) challenges” to “transportation (OR: 2.51, 95% CI: 1.03 – 6.28) challenges and food insecurity (OR: 2.56, 95% CI: 1.05– 6.43)”.

There are two periods after “challenges”. Please remove one.

Change ”on the health of vulnerable preschool” to “obese preschool”.

Response: These changes were made to the abstract.

Page 2

Comment: Change “had obesity and prevalence increased” to “had obesity with its prevalence increased”.

Response: This sentence was changed.

Comment: “It was hypothesized that families with a child with a disability would be disproportionately impacted by COVID-19 related social challenges versus families who did not have a child with a disability”—what was the hypothesis regarding obesity in pre-school aged children? This paper is also on obesity. There needs to by a hypothesis related to obesity.

Response: Thank you for the comment. A hypothesis has been added. “It was also hypothesized that families with a child who is obese would be disproportionately impacted by COVID-19 related social challenges as compared to families with a child who is normal weight.”

Page 6

Although the Table 1 has been improved, please check the Children template for tables. Notice that tables are to have only horizontal lines as dividers. There should be no vertical lines as a border or to divide columns. As well, ensure that the size of the columns are such that the titles “Overall” and “Disability” are both on one line only. Increasing their size will also ensure that the data are all only on one line a well. Under the Non-Disability column for Ethnicity, the percentages add up to 99% rather than 100%. Please change it so it adds up to 100%.

Response: Thank you for the comment. These changes have been made to Table 1.

Page 7

Change “families whom Spanish” to “families for whom Spanish”.

Change “nostatistically” to “no statistically”.

Response: We made these changes as noted.

Pages 7-9

Although the Table 1 has been improved, please check the Children template for tables. Notice that tables are to have only horizontal lines as dividers. There should be no vertical lines as a border or to divide columns. As well, ensure that the size of the table is such that the title “Overweight” is on one line only. To have the English correspond with the other titles, change “Obesity” to “Obese”.

Response: Thank you for the comments. These changes were made.

Page 9

“Also, families who were worried about transportation due to COVID-19 were over 7 times as likely to have a child with obesity versus parents who were not worried about transportation barriers”—this doesn’t seem right. Please check this result.

Response: You are correct. This was changed to “2 times”…..

Pages 9-10

Table 3 is done in the correct format according to the Children template except that the line above the subheading should be solid, not broken by columns. 

Response: We made these changes as noted.

Page 10

The footnotes for Table 3 are repeated. Eliminate one set of footnotes.

Response: The second footnotes were deleted.

“Figure 1 shows 2 models with”—please increase the font size of these words to correspond with the rest of the main text

Response: We made these changes as noted.

Page 11

Both Figure 1A and Figure 1B are far too small to read the text. Their size needs to be increased so that the text is legible.

Change the title of Figure 1B from “Outcome2: Child Who Was Obesity” to “Outcome 2: Child Who Was Obese”.

Response: We made these changes as noted.

Page 12

Comment: Change “families who had not” to “families who didn’t have”.

Response: This was revised.

Comment: “Parents who were born overseas were over 9 times more likely to not have enough money to get more food (OR 9.34, 95% CI 2.42-45.0, p- value=0.002) and over 5 times more likely to could not be able to afford balanced meal”—9 times and 5 times seem too large, is this really right?

Response: The biostatistician re-ran these statistics and both estimated odds ratio are correct based on our sample. We have 7(11%) U.S. born participants reported not have enough money to get more food compared to 48 (52%) participants who born overseas. We also have 10 (15%) U.S. born participants reported cannot afford balanced meal compared to 50 (55%) participants who born overseas. We added this to the results. Our discussion and limitations sections already take note of the imbalanced sample sizes.

Comment: Change “more likely to could not be able to afford” to “more likely to not be able to afford”.

Response: This change has been made.

Page 13

Table 4 is done in the correct format according to the Children template except that the line above the subheading should be solid, not broken by columns. 

Change, in column 1 “Child who was Obesity” to “Child who was Obese”.

Response: This change has been made.

Page 14

Comment: Both the first and second paragraphs of the Discussion have periods at the beginning of their first sentences. Please remove these.

Response: This change was made.

Comment: Change “ethnically minoritized” to “Hispanic”. Hispanic families are not in a minority in Miami.

Response: This change was made.

Change “areobese” to “are obese”.

Response: This change was made.

Change “Results suggest that families of children who areobese had more significant pandemic impacts on transportation and food insecurity challenges” to “Results suggest that there are more significant pandemic impacts on transportation challenges and food insecurity for families of children who are obese.”

Response: This sentence was rewritten as suggested. “Results suggest that there are more significant pandemic impacts on transportation challenges and food insecurity for families of children who are obese.”

Change “a preschooler who areobesity” to “a preschooler who is obese”.

Response: This change was made.

Change “[31,32].In” to “[31,32]. In”.

Response: This change was made.

Change “food market that with” to “food markets with”.

Response: This change was made.

Change “diverse racial/ethnic” to “Hispanic”.

Response: This change was made.

Page 15

“One explanation for this finding is that families with a child with a disability may have parental resiliency factors that may buffer impacts”—need a current reference with respect to resiliency for this explanation.

Response: The study cited in reference [39] and the sentence below explains resiliency in this regard.

Change “.. Study” to “Study”.

Response: This change was made.

Change “that prevalence” to “that the prevalence”.

Response: This change was made.

Change “families whom” to “families for whom”.

Response: This change was made.

Change “and specifically amoug Hispanics in general” to “specifically among Hispanics”.

Response: This change was made.

Change “that who were” to “that those who were”.

Response: This change was made.

Change “stamps -supplemental” to “stamps, supplemental”.

Response: This change was made.

Comment: One thing that is not mentioned regarding the overweight and obesity found in Hispanics that were studied is that overweight in infants may be a result of maternal excessive stress. Please add current references to stress and weight gain and suggestions regarding the possibility of counselling to decrease maternal stress. See, for example:

Carroll, N.; Sadowski, A.; Laila, A.; Hruska, V.; Nixon, M.; Ma, D.W.L.; Haines, J.; on behalf of the Guelph Family Health Study. The Impact of COVID-19 on Health Behavior, Stress, Financial and Food Security among Middle to High Income Canadian Families with Young Children. Nutrients 202012, 2352. https://doi.org/10.3390/nu12082352

Jansen, E.; Thapaliya, G.; Aghababian, A.; Sadler, J.; Smith, K.; Carnell, S. Parental stress, food parenting practices and child snack intake during the COVID-19 pandemic. Appetite 2021161, 105119. https://doi.org/10.1016/j.appet.2021.105119

Response:  Thank you for this comment and suggestion. We agree that this is an important area to consider. We need this to the discussion and to the references.

Please delete the period before the final paragraph preceding Study Limitations and Strengths.

Response: This change was made.

Change “ethnic and racial backgrounds specifically Hispanic or Non-Hispanic Black” to “Hispanic or Non-Hispanic Black”

Response: This change was made.

Change “data was” to “data were” (this error appears in the final paragraph of the page twice).

Response: This change was made.

Page 16

In the Conclusions, recommendations should also be made regarding ways to decrease the stress of those Hispanic families of overweight and obese children.

Response: A sentence was added to the conclusions regarding decreasing stress.

Comment: Please reduce the font size of “Institutional Review Board Statement”, “Informed Consent Statement”, “Data Availability Statement: Given the confidentiality of subjects, data is not available to be shared.”, “Conflicts of Interest” and “References”.

Response: The font size has been reduced.

Although the style of the references is much closer to MDPI style, none of the necessary italics or bolding has been done. Please fix all the references to include italics and bolding. As well, note that it should say “Available online:” and after the address, it should say “(accessed on Day Month written as a word and Year)”.

Response: These have been addressed.